# Maize Straw Strip Mulching as a Replacement for Plastic Film Mulching in Maize Production in a Semiarid Region

**Xuemei Lan [1,2], Shouxi Chai [1,2,*], Jeffrey A. Coulter [3] 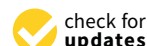, Hongbo Cheng [4], Lei Chang [1,2], Caixia Huang [5], Rui Li [1,2], Yuwei Chai [1,2], Yawei Li [1,2], Jiantao Ma [1,2] and Li Li [1,2]**

1   Gansu Provincial Key Laboratory of Aridland Crop Science, Lanzhou 730070, China; xuemei.lan@outlook.com (X.L.); cl593571@outlook.com (L.C.); lirui20200802@outlook.com (R.L.); cywwy@outlook.com (Y.C.); Liyaweikl@outlook.com (Y.L.); mjt0323@outlook.com (J.M.); lily200802@outlook.com (L.L.)
2   College of Agronomy, Gansu Agricultural University, Lanzhou 730070, China
3   Department of Agronomy and Plant Genetics, University of Minnesota, St. Paul, MN 55108, USA; jeffcoulter@umn.edu
4   College of Life Science and Technology, Gansu Agricultural University, Lanzhou 730070, China; chb0802@outlook.com
5   College of Water Conservancy and Hydropower Engineering, Gansu Agricultural University, Lanzhou 730070, China; hcx0801@outlook.com
*   Correspondence: chaisx@gsau.edu.cn; Tel.: +86-138-9335-7084

**Abstract:** Straw strip mulching in maize (*Zea mays* L.) production is showing a positive effect with the increasing negative effects coming from crop straw and plastic film residues. Therefore, it is imperative to develop comprehensive utilization of straw, and promote the green development of agriculture in rainfed regions. A dryland field experiment was conducted in semiarid northwestern China in 2017 and 2018 and included three treatments: maize straw strip mulching with alternating strips of mulched and non-mulched soil (MSSM), and double ridge-furrow fully mulched soil with white or black plastic film (DRWP or DRBP, respectively). The results show that the interaction between mulching treatment and year significantly influenced maize silage yield, grain yield, biomass yield, aboveground plant water content at silage maize harvest stage, ears ha$^{-1}$, kernels ear$^{-1}$, and thousand kernel weight ($p < 0.001$, $p = 0.002$, $p < 0.001$, $p < 0.001$, $p < 0.001$, $p < 0.001$, $p < 0.001$, and $p < 0.001$, respectively). For silage, maize growth under straw strip mulching was greater than that of the double ridge mulching system. Silage yield of MSSM was significantly higher than that of DRWP and DRBP, but maize grain and biomass yields under MSSM were significantly lower those under DRWP and DRBP in 2017 and 2018. Compared with the double ridge mulching system, net economic return from silage with MSSM was significantly increased by 28.31% and 20.85% in 2017 and 2018, respectively, and net economic return from grain was 6.67% lower in 2017 and 2.34% higher in 2018. The MSSM treatment exhibited water-temperature coupling; the MSSM treatment significantly reduced soil temperature in the 0–25 cm soil layer by 1.23–2.14 °C and increased soil water storage in the 0–200 cm soil layer by 9.75 and 24.10 mm in 2017 and 2018, respectively, thereby delaying growth development of maize by about 13 days. Therefore, straw mulch can replace plastic film mulch and serve as an environmentally friendly cultivation method for maize in semiarid rainfed regions.

**Keywords:** straw strip mulching; double ridge-furrow system with plastic mulching; soil temperature; ecological economy benefits; dryland

## 1. Introduction

Maize (*Zea mays* L.), wheat (*Triticum aestivum* L.), and rice (*Oryza sativa* L.) are the three most widely grown crops in the world, and over 30% of human caloric intake comes from maize in 94 developing countries [1,2]. The land under cultivation has increased dramatically over the past 50 years, with more than 0.1 billion hectares of maize produced in at least half of all developing countries [1,2]. Maize also plays a key role in silage and grain for animal feed, and is used widely in industrial products all over the world [2]. With increasing demands and yield scarcity in world maize supplies, food insecurity is worsening, particularly for poorer consumers. Therefore, farming practices ensuring high and stable yields are important to producers in many developing countries, including in the tropics, Africa, Southern Africa, the East African highlands, Ethiopia, Tanzania, Uganda, and in China's Loess Plateau, to meet future maize demands, especially in semiarid regions. In the Loess Plateau of northwestern China [3,4], the maize planting area in Gansu Province has increased by 96% from 2006 to 2015, and Gansu has the highest area and total production of maize among the five northwestern provinces of China [4]. However, the primary constraints on crop production in this region are degraded soils, inefficient water use, drought, and low temperature stress early in the growing season [5–7], which severely restrict the profitability of rainfed crop production for local farmers [8,9]. To solve these issues, many management strategies have been studied to improve rainwater use efficiency (WUE), and collecting effective precipitation during the crop growing season is key to improving crop yield [10]. Currently, the ridge-furrow system [11] and plastic film mulching [9,12] are effective methods for increasing rainfed maize productivity in semiarid environments with low air temperature in the spring, due to their ability to increase soil temperature, reduce evaporation of effective precipitation, increase soil water content, and prolong the reproductive growth cycle [13–15]. Plastic film as a mulching material to cover the soil surface has brought enormous economic benefits to farmers in semiarid environments due to improved soil hydrothermal conditions for crop growth and yield, but use of plastic film has negative effects on the environment [16–18] and contributes to soil pollution [19]. Residual plastic film is difficult to degrade and its recycling capacity is inefficient, which is a major limiting factor for sustainable development of agriculture; it reduces the homogeneity of soil, and it can seriously impede the movement of water and solutes in soil [18,20].

However, a larger negative of plastic mulching on the soil, air, and environment relates to how it might impact human life in the future [21]. Meanwhile, crop straw residue is on the rise with the increasing demand for food security, and burning has become one of the shortcut ways to deal with straw residue before the new cropping seasons. Rice production in India, Thailand, and the Philippines are examples [22]. When maize is harvested for grain, less than 10% of the crop straw on the land returns to the soil, leading to straw being burned on the land before new crops are produced, which has caused serious environmental pollution in the last decade [23]. These agricultural residues have a negative impact on the sustainability of food security as their disposal is a major environmental and economic impact issue [24]. In Asia, open field burning (including living and dead vegetation) annually burns about 730 Tg of biomass [22]. Air pollution is an important environmental issues and is currently one of the greatest threats to human health in China [23]. The development of new mulching technologies can promote closer cooperation between decision makers and stakeholders [24]. Whole plant corn silage is widely used in ruminant production worldwide due to its high yield, nutritional value, and digestibility, palatability, and ease of preservation [25,26], and an analysis of energy use and economic return found that corn silage productivity value is higher on larger farms in Iran [27]. Nowadays, society is paying more attention to the consequences of crop production, not only from the perspective of economics, but also from the aspects of society, ethics, and the environment. Within the framework of the modern economy, all business activities should be carried out in accordance with the principle of sustainable development [24]. In 2017, China's Ministry of Agriculture issued the implementation plan for grain to feed, which vigorously promotes silage corn planting and technology. Hence, it is important to analyze the potential to combine grain maize production and silage maize production in sustainable agriculture and animal consumption in the future.

Recently, maize straw has been used as a mulching material instead of plastic film, which can improve the plant growth, conserve soil moisture, and return nutrients to the soil, thereby promoting sustainable development of agriculture [28–30]. Some studies have found that appropriate straw mulching methods can have positive effects [31–34]. Mulching soil with maize straw avoids soil pollution caused by plastic film mulching and reduces air pollution and soil organic carbon mineralization caused by the burning of maize straw, and utilizes the abundant straw resource for soil improvement compared to no straw retention [35–37]. Additionally, straw mulching can increase infiltration of rainfall [38], inhibit soil evaporation, improve soil moisture, modify soil temperature [39–41], enhance soil aggregation, promote soil biological activity [42,43], stimulate plant growth and development [44], and improve crop productivity and quality [45]. However, some studies have shown that straw mulching does not increase crop yield and can even result in a yield decrease [13,46,47]. This may be related to reductions in soil temperature and crop WUE under traditional straw mulching over the entire soil surface [48], which can also impede field operations such as sowing and harvesting [49]. Therefore, to solve the conflict between soil temperature reduction and soil moisture conservation that occurs when straw mulching is applied to the whole soil surface [48], our team developed a new straw mulching technique [50], in which straw mulch is applied in strips and wheat (*Triticum aestivum* L.) or potato (*Solanum tuberosum* L.) is planted in the non-mulched strips [51–53]. This technique requires a smaller quantity of straw compared to mulching the entire soil surface, making it easier for smallholder farmers to implement since they remove straw from the field and chop it into smaller pieces before returning it to the field [50]. Straw strip mulching can restrain the evaporation of surface water, maintain soil moisture, and adjust soil temperature, thereby improving the yield of rainfed wheat and potato in semiarid regions compared to no mulching and plastic film mulching. Our previous studies showed that the straw strip mulching significantly advanced winter wheat yield by 11.9–19.5% and decreased soil temperature by 0.7–1.3 °C compared with traditional non-mulched flat planting, and it increased potato yield by 36.9–61.2% and enhanced water use efficiency by up to 74.8% [51–53].

Previous research has investigated the mechanisms of wheat and potato yield improvement under straw strip mulching, but to our knowledge no information has reported the effects of soil temperature and moisture under maize production with straw strip mulching in rainfed semiarid regions. In this study, we conducted a two-year field experiment to explore the interactive effects of straw strip mulching on grain and silage maize production in a rainfed semiarid area. Maize straw strip mulching (MSSM) with alternating strips of mulched and non-mulched soil, double ridge-furrow fully mulched soil with white or black plastic film (DRWP or DRBP, respectively), with the DRWP treatment serving as the control treatment, were compared to determine their effects on soil temperature and maize silage, grain, and biomass yields. The objectives of this study were to: (a) compare the effects of DRWP, DRBP, and MSSM on soil temperature and soil water storage (SWS); (b) evaluate the effects of DRWP, DRBP, and MSSM on maize growth development accumulation at different stages, silage yield, grain yield and biomass yield; (c) assess the economic performance of straw strip mulching compared with plastic mulching; and (d) seek the influence of the sustainable agricultural and ecological environment effects before and after straw strip-mulched crops in rainfed agricultural regions.

## 2. Materials and Methods

### 2.1. Site Description

A field experiment was conducted under rainfed conditions in 2017 and 2018 at the Tongwei Modern Dryland Circular Farming Experiment Station in Dingxi, Gansu Province, China (35°11′ N, 105°19′ E; altitude 1740 m). The study site is characterized as a typical semiarid rainfed agricultural region in the Loess Plateau, with one crop produced in each year. According to long-term (1975–2015) meteorological data from the study site, annual mean air temperature is 7.2 °C, annual mean sunshine is 2092 h, the frost-free period is 155 d, and annual mean precipitation is 390.6 mm (250–550 mm),

which occurs mostly from June through September, whereas annual mean soil evaporation is 1500 mm. The highest values of mean monthly air temperature and precipitation were recorded in July and August, respectively, during 1975–2015. The soil at the experimental site is a loessal soil [54], which has a sandy loam texture with ≥50% sand, and is locally referred to as Huangmian sandy loam [55], with an average bulk density of 1.25 g cm$^{-3}$ in the 0–200 cm layer. Total N, nitrate-N, ammonium-N, available phosphorous, available potassium, and PH in the 0–20 cm layer at the start of the experiment were 0.87 g kg$^{-1}$, 14.5 mg kg$^{-1}$, 2.7 mg kg$^{-1}$, 10.7 mg kg$^{-1}$, 99.9 mg kg$^{-1}$, and 8.2, respectively. Water holding capacity and permanent wilting point for the 0–200 cm soil layer is 6.9% and 24.8%, respectively [52].

## 2.2. Experimental Design and Field Management

On 12 March 2017 and again on 5 November 2017, a total of 180 kg nitrogen (N) ha$^{-1}$ and 150 kg $P_2O_5$ ha$^{-1}$ were uniformly applied to the soil surface of the entire experimental site, using urea (46% N) and triple superphosphate (46% $P_2O_5$ and 8% N). The soil was moldboard plowed to a depth of 20 cm, rotary tilled to a depth of 10 cm, firmed, and mulched with plastic film or maize straw in the day of after fertilizer management. The next step involved application of the mulching treatments. The three mulching treatments for maize production (DRWP, DRBP, and MSSM) were arranged in a randomized complete block design with three replications in plots that were 4.8 m wide and 60 m long, and were applied on 15 March of 2017 and 8 November of 2017. The treatments were applied to the same plots for both growing seasons. The DRWP and DRBP treatments had alternating narrow (0.4 m wide and 0.1 m high) and wide (0.7 m wide and 0.1 m high) ridges that were both covered with polyethylene film (0.01 mm diameter, 120 cm width). Small holes (20 mm diameter) were made in the plastic film at 0.4 m intervals in the furrows with a sharp object for rainfall infiltration. In the DRWP and DRBP treatments, two rows of maize were planted 0.6 m apart in the wide ridges with 0.6 m spacing between rows. The MSSM treatment had alternating strips of mulched (0.5 m wide) and non-mulched (0.7 m wide) soil and two rows of maize were planted 0.6 m apart in the non-mulched strips (Figure Graphical Abstract). For the MSSM treatment, maize straw was collected following grain harvest, air-dried, and then uniformly applied to strips without additional chopping into smaller pieces at a rate of 9.0 t ha$^{-1}$, which is equivalent to the rate of local maize straw production.

Maize (cv. Dunyu12) was directly planted into the soil of each plot by a hand-held maize planting machine on 25 April 2017 and 17 April 2018 to a target density of 55,500 plants ha$^{-1}$. Half of the plants from each plot were harvested for measurement of silage yield on 15 September 2017 and 08 September 2018, and the remaining 50% of the plants from each plot were harvested for grain yield on 21 October 2017 and 10 October 2018. Maize management was consistent with the practices of local farmers, and included no irrigation and removal of weeds by hand.

## 2.3. Measurements and Calculations

### 2.3.1. Soil Temperature

The dates of the following maize growth stages were recorded when 75% of the plants were at these stages, according to [56]: emergence (VE), two leaf collar (V2), six leaf collar (V6), 10 leaf collar (V10), tasseling (VT), silking (R1), blister (R2), milk (R3), and physiological maturity (R6). In each plot, curved pipe geothermometers (Hongxing Thermal Instruments, Wuqiang County, Hebei Province, China) were placed in the soil at 5, 10, 15, 20, and 25 cm depths between two maize plants in a row in the central area of the plot [57]. Soil temperature was measured at 7:00, 14:00, and 19:00 h, and the average value of the three readings was used as the daily average soil temperature. Effective accumulated soil temperature in each growth period of maize was calculated as the product of the average soil temperature (≥5 °C) of the 5–25 cm soil layer in a growth period and the duration of growth period.

### 2.3.2. Soil Water Storage

Each plot was surrounded by 20 cm high ridges to prevent the occurrence of surface runoff from precipitation during this study. The groundwater level was about 50 m below the soil surface, so upward water flow to roots was considered negligible [52].

On sunny days at the VE, V2, V6, V10, R1, R3, and R6 stages of maize, soil gravimetric water content was measured from the 0–20, 21–40, 41–60, 61–90, 91–120, 121–150, 151–180, and 181–200 cm soil layers in each plot. Soil samples were taken using a 5 cm diameter handheld soil auger between two plants in a row from three random sampling points near the center of each plot. Soil gravimetric water content was determined by the oven-drying method at 105 °C [58]. If rainfall occurred on a planned sampling day, soil sampling was done 2–3 days later. The mean 0–200 cm soil gravimetric water content was calculated as the weighted average of the eight soil layers as follows:

$$ \text{GWC} = \frac{\sum_1^i \text{GWC}_i \text{SD}_i}{\sum_1^i \text{SD}_i} \tag{1} $$

where GWC is the weighted average soil gravimetric water content (%), $\text{GWC}_i$ is gravimetric water content for a given soil layer, and $\text{SD}_i$ is the soil depth (cm) for a given soil layer. Soil water storage was determined by the following equation [57,58]:

$$ \text{SWS (mm)} = \text{GWC (\%)} \times \rho_b \left( \text{g cm}^{-3} \right) \times \text{SD (cm)} \tag{2} $$

where ρb is soil bulk density, with an average bulk density is 1.25 g cm$^{-3}$ in the 0–200 cm layer [52], and SD is soil depth.

### 2.3.3. Aboveground Fresh Yield

The aboveground fresh yield was measured by harvesting and weighing five randomly selected plants in each plot at the 72 days (V10), 108 days (VT), 140 days (R3) and 176 days (R6) after planting.

### 2.3.4. Silage, Grain, and Biomass Yield

Total aboveground fresh yield of maize was measured by harvesting and weighing 50% of the plants in each plot when maize was at the mid-dent (50% kernel milk line) stage (R3) [59]. Grain yield components and total aboveground biomass (i.e., biomass yield) of maize were measured at R6 from 30 randomly selected plants in each plot, while grain yield of maize was measured by harvesting the remaining plants in each plot at R6.

### 2.3.5. Economic Benefits

Analyses of the economic benefits of maize silage and grain under the different mulching practices were based on total investment cost and revenue. Revenue from silage corn (Chinese Yuan ha$^{-1}$) was calculated as the product of aboveground fresh yield at the mid-dent stage of maize and a silage corn price of 260 Chinese Yuan per 1000 kg$^{-1}$, based on price estimates from the local area. Revenue from grain (Chinese Yuan ha$^{-1}$) was calculated as the product of maize grain yield and grain price obtained from the China National Agricultural Market Service Database [60]. Maize grain price averaged 1.8 Chinese Yuan per kg$^{-1}$. Net economic return (Chinese Yuan ha$^{-1}$) was calculated separately for silage and grain, as revenue from silage or grain minus total investment cost, which included seed, pesticides, and chemical fertilizers (1866 Chinese Yuan ha$^{-1}$), cost of plastic film (3150 Chinese Yuan ha$^{-1}$ for both black and white plastic film), and mechanical field operations and labor costs for fertilizer application, tillage, mulch preparation and application, and planting (3000 Chinese Yuan ha$^{-1}$). No cost

was attributed to the maize straw for the MSSM treatment. The ratio of total investment cost to revenue from silage was calculated as an index of silage maize economic benefit as follows:

$$\frac{\text{Total investment cost}}{\text{Silage yield } \times \text{ selling price}} \tag{3}$$

The ratio of total investment cost to revenue from grain was calculated as an index of grain maize economic benefit as follows:

$$\frac{\text{Total investment cost}}{\text{Grain yield } \times \text{ selling price}} \tag{4}$$

### 2.4. Statistical Analyses

Data were analyzed using Genstat edition 20 software (VSN Intl. Ltd., Oxford, UK). Analysis of variance was conducted for all dependent variables, and means were compared using Fisher's protected least significant difference test. Treatment and year were considered fixed effects, while replication and interactions with replication were considered random effects.

## 3. Results

### 3.1. Weather Conditions

The mean, maximum, and minimum air temperatures from April through October were 15.0, 32.0, and −3.0 °C in 2017, and 15.0, 29.0, and −5.0 °C in 2018, respectively. Precipitation during the maize growing season in the 40 years (1975–2015) prior to this study averaged 213 mm, while that in 2017 and 2018 was 273 and 365 mm, respectively (Figure 1). The total and effective precipitation (>5 mm) during the maize growing season was 334 and 273 mm in 2017, and 420 and 365 mm in 2018, respectively. The effective precipitation amounts and timings, and frequency of precipitation events >15 mm were greater in 2018 than in 2017. The greatest amount of precipitation occurred between the V10 and R1 stages, an intermediate amount occurred between the V2 and V6 stages, and the least amount occurred between the R3 and R6 stages. In 2017 and 2018, the effective precipitation (and proportion to the whole growing period) was 21.4 mm (5.5%) and 41.9 mm (8.6%) from the VE to V2 stages, 65.7 mm (16.7%) and 28.3 mm (5.8%) from the V3 to V6 stages, 48.0 mm (10.7%) and 72.7 mm (15.0%) from the V7 to V10 stages, 110.0 mm (28.0%) and 160.1 mm (33.1%) from the VT to R1 stages, 5.5 mm (1.4%) and 40.3 mm (8.3%) from the R2 to R3 stages, and 28.6 mm (7.3%) and 20.9 mm (4.3%) from the R4 to R6 stages, respectively.

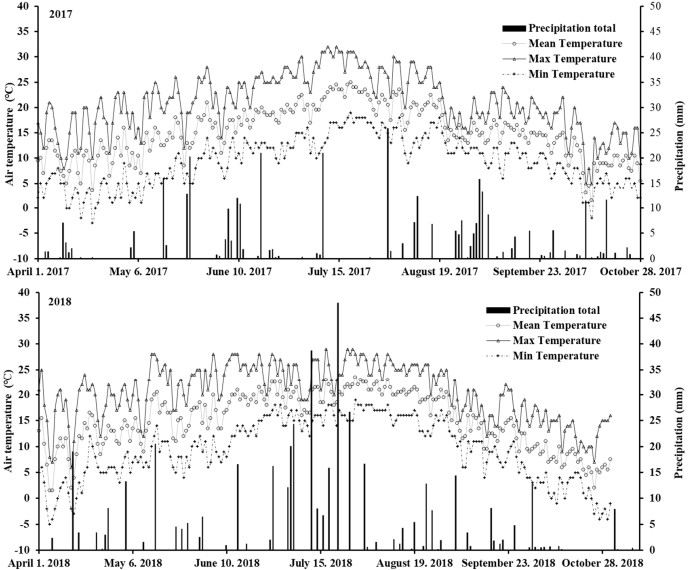

**Figure 1.** Daily precipitation and mean, maximum, and minimum air temperatures during the maize growing season at Tongwei, Gansu Province, China in 2017 and 2018.

### 3.2. Soil Temperature

#### 3.2.1. Soil Temperature in the 5–25 cm Soil Layer Throughout the Growing Season

Mean soil temperature in the 5–25 cm soil layer throughout the growing season of maize differed significantly among mulching methods in the two years (Figure 2). Compared with the DRWP treatment, the DRBP and MSSM treatments significantly reduced mean soil temperature of the 5–25 cm soil layer by 0.74 and 1.96 °C in 2017, and by 0.74 and 2.14 °C in 2018 ($p \leq 0.05$).

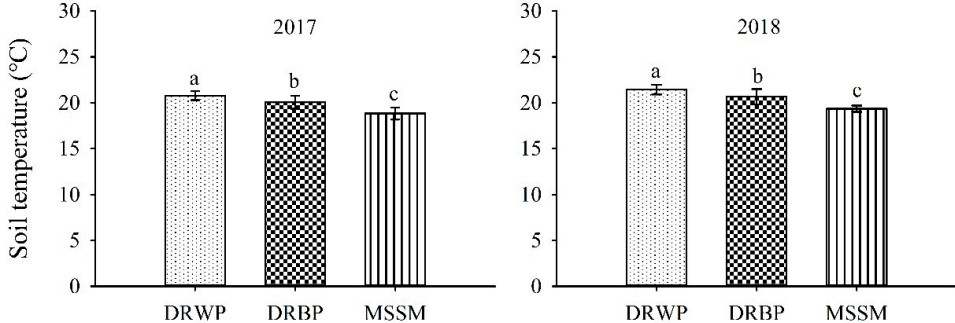

**Figure 2.** Mean soil temperature of the 5–25 cm soil layer throughout the growing season as affected by mulching method in 2017 and 2018. Within a year, different letters denote means that are significantly different ($p \leq 0.05$). Error bars indicate standard errors of the means ($n = 3$). DRWP, double ridge-furrow fully mulched soil with white plastic film; DRBP, double ridge-furrow fully mulched soil with black plastic film; MSSM, maize straw strip mulching with alternating strips of mulched and non-mulched soil and maize planted in non-mulched strips.

#### 3.2.2. Soil Temperature of Different Soil Layer Throughout the Growing Season

Mean soil temperature at the different soil layers throughout the growing season of maize differed significantly among mulching methods in the two years (Figure 3). The temperature of each soil layer decreased with the deepening of the soil layer throughout the growing season; the MSSM treatment had the least fluctuation, and the temperature of each soil layer under the MSSM treatment was significantly lower than that under the DRWP and DRBP treatments. Compared with the temperature of each soil layer under the DRWP treatment, the largest soil temperature difference between the MSSM and DRBP treatments was in the 25 cm soil layer in 2017, and was reduced by 2.57 and 1.54 °C, respectively. The largest soil temperature difference between the MSSM and DRBP treatments in 2018 was in the 5 and 25 cm soil layers, which was 2.75 and 1.73 °C lower than that of the DRWP treatment, respectively.

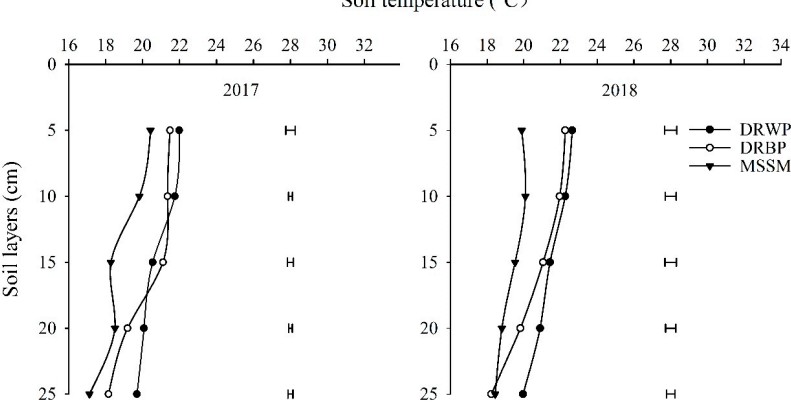

**Figure 3.** Mean soil temperature at different soil layers throughout the growing season as affected by mulching method in 2017 and 2018. Vertical bars indicate Fisher's protected least significant ($p \leq 0.05$) difference among treatments at a given stage of maize phenological development.

### 3.2.3. Soil Temperature in the 5–25 cm Soil Layer at Different Growth Stages

Mean soil temperature in the 5–25 cm layer during different growth periods of maize was significantly different among mulching methods in the two years (Figure 4). The coefficient of variation (CV) of mean soil temperature in the 5–25 cm layer during the vegetative growth period was larger in 2017 than 2018 due to the mulching time, which for the plastic film on double ridge and maize straw strip mulch on furrow for the treatments in 2017 was later than that in 2018. Compared to the DRWP treatment in 2017, mean soil temperature of the 5–25 cm soil layer with the MSSM treatment was significantly decreased from the V2 to V6 stages by 5.32–6.62 °C, and significantly increased from the R3 to R6 stages by 0.59–1.16 °C, and with the DRBP treatment was significantly decreased from the V2 to V6 stages by 2.35–2.52 °C and significantly increased from the R3 to R6 stages by 0.50–0.50 °C. Compared to the DRWP treatment in the wetter year of 2018, mean soil temperature in the 5–25 cm depth with the DRBP and MSSM treatments was significantly reduced from the V2 to R6 stages by 0.35–1.36 and 0.76–4.21 °C, respectively.

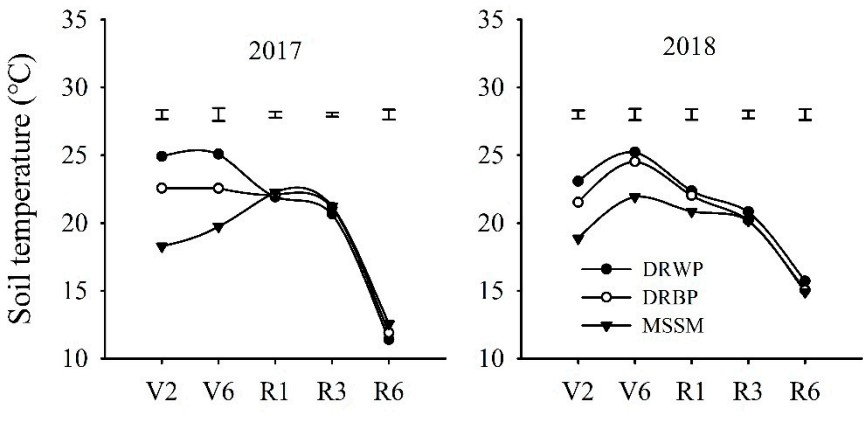

**Figure 4.** Mean soil temperature in the 5–25 cm soil layer at different growth stages as affected by mulching method in 2017 and 2018. Vertical bars indicate Fisher's protected least significant ($p \leq 0.05$) difference among treatments at a given stage of maize phenological development.

### 3.2.4. Spatiotemporal Dynamic Temperature Difference at Different Soil Layers of Different Growth Stages

Soil temperature of different soil layers in different growth periods of maize was significantly different among mulching methods in the two years (Figure 5). In 2017, the CV of the 5–25 cm soil layers in the DRWP, DRBP and MSSM treatments were highest during the vegetative growth period, and the CV values of DRBP were the lowest in other periods. In 2018, the CV values of the 5–25 cm soil layers were lowest in the MSSM treatment, indicating that straw strip mulching can reduce fluctuation of soil temperature. Compared with the temperature of different soil layers in different growth periods under the DRWP treatment, the largest soil temperature difference appeared at the V2 stage in 2017 and 2018, and soil temperature at the 15 cm soil layer of the V2 stage in 2017 decreased by 7.45 °C and at the 10 cm soil layer of the V2 stage in 2018 decreased by 4.82 °C, respectively. Among the 25 soil temperature measured points in total at the each soil layer at the V2, V6, R1, R3 and R6 stages, when compared to the DRWP treatment, the DRBP and MSSM treatments resulted in cooler soil temperatures at 48% and 44% of the measured points in 2017, and 88% and 100% of the measured points in 2018, respectively. In 2017–2018, the MSSM treatment at the R1 and R3 stages had the smallest fluctuation among soil layers, with CV values of 3.68 and 2.67, respectively. Among soil layers, the 15 cm soil layer at the R3 stage had the smallest fluctuation, with CV values of 0.60–0.76.

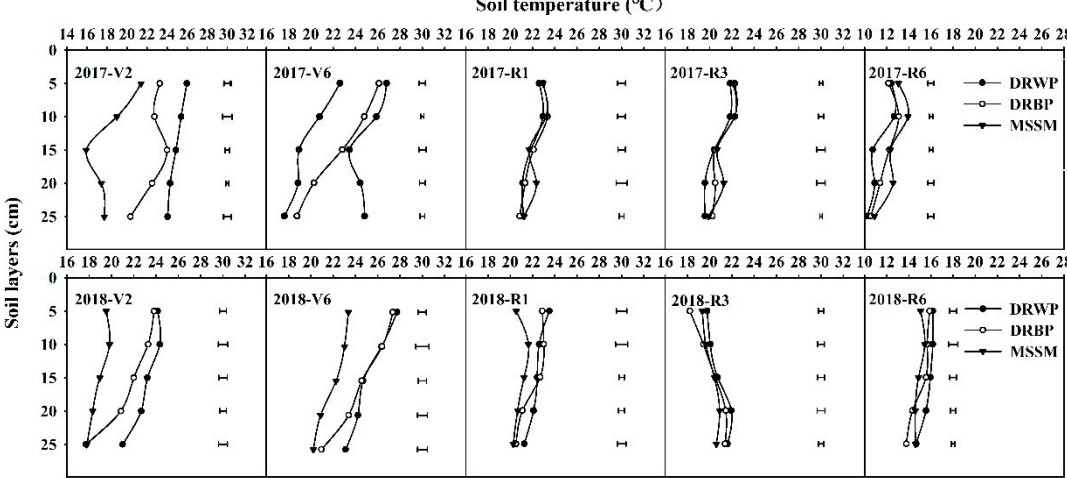

**Figure 5.** Mean soil temperature in different soil layers at different growth stages as affected by mulching method in 2017 and 2018. Vertical bars indicate Fisher's protected least significant ($p \leq 0.05$) difference among treatments at a given stage of maize phenological development.

### 3.3. Soil Water Storage in the 0–200 cm Soil Layer throughout the Growing Season

In both years, SWS in the 0–200 cm soil layer throughout the growing season was greatest with the MSSM treatment, followed by the DRBP treatment, and was least with the DRWP treatment (Figure S1). Compared with the DRWP treatment, the DRBP and MSSM treatments significantly increased mean SWS in the 0–200 cm soil layer by 14.35 and 24.10 mm in 2017, and by 4.21 and 20.00 mm in 2018, respectively.

### 3.4. Maize Growth Development Accumulation at Different Stages

#### 3.4.1. Maize Growth Day Accumulation at Different Stages

After planting, the growth day accumulation of maize was shortened with increased with effective precipitation, and it was significantly different among mulching methods in the two years (Figure S2). The DRWP, DRBP, and MSSM treatments took 165, 166 and 179 days to reach R6 in 2017, and 161, 163, and 176 days to reach R6 in 2018, respectively. There were no differences in growth day accumulation between the DRBP and DRWP treatments, but the MSSM treatment significantly delayed time from planting to R6 by 13–15 days compared to the DRWP and DRBP treatments in 2017 and 2018. (Figure S2). Compared with the DRWP treatment, the V2, V6, R2, and R6 stages of MSSM treatment were delayed by 7, 3, 3, and 3 days in 2017, and by 7, 2, 3, and 3 days in 2018, respectively.

#### 3.4.2. Accumulated Temperature at Different Stages

The mean daily temperature accumulated after planting under the different mulching treatments was significantly different at different growth stages and throughout the whole growth stages across the two years ($p < 0.05$) (Figure S3). Total mean daily temperature accumulation increased with maize growth day accumulation throughout the whole growth period, and total accumulated daily soil temperature under the MSSM treatment was significantly greater than that under the DRWP and DRBP treatments across the two years. Compared with the total accumulated daily soil temperature under the DRWP treatment in 2017, the DRBP and MSSM treatment were significantly decreased and increased by 22.72 and 118.06 °C, respectively. In 2018, the total accumulated daily soil temperature under the MSSM treatment was not significantly different with the DRWP treatment; in contrast, that of the DRBP treatment was 61.35 °C lower when compared to the DRWP treatment.

Compared with the DRWP treatment, the mean daily temperature accumulated with the DRBP and MSSM treatments showed significant warming and cooling effects with maize growth day

accumulation in 2017 and 2018 (Figure S3). Compared to the DRWP treatment, mean daily temperature accumulated under the DRBP treatment in 2017 was increased by 14.62 and 23.00 °C at the V2 and R6 stages, and decreased by 38.01, 10.55, and 11.78 °C at the V6, R1 and R3 stages, and that of DRBP treatment in 2018 was increased by 18.34 and 7.13 °C at V2 and R6 stages, and decreased by 42.28, 31.4, and 13.14 °C at R1, R3 and R6 stages, respectively. Compared to the DRWP treatment, mean daily temperature accumulated under the MSSM at the V2, R3 and R6 stages was increased by 41.79, 74.97, and 91.22 °C in 2017, and by 64.96, 49.01, and 10.92 and 2018, respectively, and at the V6 and R1 stages was decreased by 68.37 and 21.55 °C in 2017 and by 38.11 and 88.98 °C in 2018, respectively.

### 3.4.3. Aboveground Fresh Weight Accumulation

Mean aboveground fresh weight accumulation per plant after maize planting was significantly different among mulching methods in the two years (Figure S4). Aboveground fresh weight accumulation of the DRBP treatment was lower than the DRWP treatment throughout the whole growth period, and that of the MSSM treatment was lower than the DRWP and DRBP treatments from 0 to 134 days, but fresh weight accumulation was greater after 135 days. Across the two years, when compared to the DRWP treatment, aboveground fresh weight per plant with the MSSM treatment at 73, 92–94, and 128–134 days after planting was significantly decreased by 47.98–75.71%, 28.59–29.60%, and 4.81–5.30%, respectively; in contrast, at 143–144 and 149–151 days after planting, it was significantly increased by 0.09–4.89% and 6.76–8.42%, respectively.

### 3.5. Silage, Grain, and Biomass Yield, Silage Harvest Parameters, and Grain Yield Components

The interaction between mulching treatment and year significantly influenced maize silage yield, grain yield, biomass yield, aboveground plant water content at silage maize harvest, ears per hectare, kernels per ear, and thousand kernel weight ($p < 0.001$, $p = 0.002$, $p < 0.001$, $p < 0.001$, $p < 0.001$, $p < 0.001$, $p < 0.001$, and $p < 0.001$, respectively). Silage yield and aboveground plant water content at silage maize harvest of the MSSM treatment was significantly higher than that of the DRWP and DRBP treatments, but maize grain yield, biomass yield, and grain yield components under MSSM were significantly lower those under DRWP and DRBP in 2017 and 2018 (Table 1).

**Table 1.** Silage, grain, and biomass yields, silage harvest parameter, and grain yield components as affected by mulching method in 2017 and 2018.

| Years | Mulching method [a] | Yield (kg ha$^{-1}$) | | | Silage harvest parameter | | | Grain yield components | | |
|---|---|---|---|---|---|---|---|---|---|---|
| | | Silage | Grain | Biomass | $X_1$ | $X_2$ | $X_3$ | $X_4$ | $X_5$ | $X_6$ |
| 2017 | DRWP | 105,467b [b] | 9873a | 23,806a | B | 1 | 75.1c | 53,043a | 553b | 337a |
| | DRBP | 102,188c | 9793b | 22,633b | B | 1 | 76.6b | 52,255b | 571a | 328b |
| | MSSM | 112,376a | 7724c | 19,410c | A | 2 | 82.0a | 52,360b | 531c | 277c |
| 2018 | DRWP | 129,938b | 11,551a | 28,440a | B | 1 | 75.0c | 53,130a | 659a | 347a |
| | DRBP | 124,722c | 11,081b | 27,730b | B | 1 | 76.8b | 52,360b | 666a | 333b |
| | MSSM | 135,334a | 9726c | 26,593c | A | 2 | 79.0a | 51,853c | 640b | 308c |
| | Mulching method (M) | <0.001 | <0.001 | <0.001 | — | — | <0.001 | <0.001 | <0.001 | <0.001 |
| $p > F$ | Years (Y) | <0.001 | <0.001 | <0.001 | — | — | <0.001 | 0.009 | <0.001 | <0.001 |
| | M * Y | <0.001 | 0.002 | <0.001 | — | — | <0.001 | <0.001 | <0.001 | <0.001 |

[a] DRWP, double ridge-furrow fully mulched soil with white plastic film; DRBP, double ridge-furrow fully mulched soil with black plastic film; MSSM, maize straw strip mulching with alternating strips of mulched and non-mulched soil and maize planted in non-mulched strips; silage maize harvest parameters: X1, stay-green of the whole plant (A, >93%; B, >90–93%); X2, ears at or beyond the 50% kernel milk-line (mid-dent) stage (1, >75%; 2, >70%); X3, aboveground plant water content at silage maize harvest (%); grain yield components (X4, ears per hectare; X5, kernels per ear; X6, thousand kernel weight (g)). [b] Within a row for a given year, means followed by different letters are significantly different ($p \leq 0.05$).

For silage production, maize growth development under straw strip mulching was greater than that of the double ridge mulching system. Silage yield of the DRBP treatment was significantly reduced by 3.11% and 4.01%, when compared to that of the DRWP treatment; in contrast, that of the MSSM treatment was significantly increased by 6.55% and 4.15% in 2017 and 2018 (Table 1). Across the

two study years, silage yield was significantly positively correlated with mean SWS in the 0–200 cm throughout the maize growing season, but silage yield was significantly negatively correlated with mean soil temperature in the 0–25 cm depth layer and soil temperature at the 5, 10, and 15 cm depths throughout the maize growing season (Table 2). Additionally, soil temperature at the 5, 10, 15, 20, 25 cm soil depths at the V2 stage and soil temperature at the 5, 10, and 15 cm soil depths at the V6 stage were significantly negatively correlated with silage yield. In contrast, soil temperature at the 20 cm soil depth at R1 and soil temperature at the 5, 10, and 20 cm soil depths at R6 were significantly positively correlated with silage yield (Table 2).

**Table 2.** Pearson's correlation coefficients for correlations of maize silage, grain, and biomass yield with soil water storage (SWS) and mean daily soil temperature (ST) in different soil layers at different maize growth stages, across mulching methods in 2017 and 2018.

| Variable [a] | Soil Depth (cm) | 2017 Yield (kg ha$^{-1}$) | | | 2018 Yield (kg ha$^{-1}$) | | |
|---|---|---|---|---|---|---|---|
| | | Silage | Grain | Biomass | Silage | Grain | Biomass |
| SWS (MGP) | 0–200 | 0.572 | −0.823 ** [b] | −0.918 ** | 0.893 ** | −0.971 ** | −0.955 ** |
| ST (MGP) | 0–25 | −0.757 * | 0.939 ** | 0.975 ** | −0.804 ** | 0.981 ** | 0.986 ** |
| Mean ST throughout the whole growth period | 5 | −0.760 * | 0.911 ** | 0.914 ** | −0.919 ** | 0.964 ** | 0.938 ** |
| | 10 | −0.860 ** | 0.979 ** | 0.974 ** | −0.902 ** | 0.989 ** | 0.958 ** |
| | 15 | −0.866 ** | 0.981 ** | 0.977 ** | −0.901 ** | 0.964 ** | 0.947 ** |
| | 20 | −0.600 | 0.840 ** | 0.919 ** | −0.684 * | 0.933 ** | 0.963 ** |
| | 25 | −0.566 | 0.820 ** | 0.916 ** | −0.156 | 0.611 | 0.708 * |
| Mean ST at the two leaf collar stage | 5 | −0.712 * | 0.911 ** | 0.962 ** | −0.932 ** | 0.980 ** | 0.947 ** |
| | 10 | −0.792 * | 0.951 ** | 0.979 ** | −0.882 ** | 0.988 ** | 0.974 ** |
| | 15 | −0.820 ** | 0.966 ** | 0.984 ** | −0.853 ** | 0.957 ** | 0.955 ** |
| | 20 | −0.821 ** | 0.968 ** | 0.979 ** | −0.759 * | 0.948 ** | 0.965 ** |
| | 25 | −0.689 * | 0.901 ** | 0.957 ** | −0.233 | 0.680 * | 0.778 * |
| Mean ST at the six leaf collar stage | 5 | −0.875 ** | 0.982 ** | 0.962 ** | −0.934 ** | 0.936 ** | 0.907 ** |
| | 10 | −0.862 ** | 0.983 ** | 0.980 ** | −0.931 ** | 0.958 ** | 0.905 ** |
| | 15 | −0.781 * | 0.950 ** | 0.983 ** | −0.917 ** | 0.928 ** | 0.902 ** |
| | 20 | −0.439 | 0.719 * | 0.843 ** | −0.859 ** | 0.948 ** | 0.931 ** |
| | 25 | −0.379 | 0.674 * | 0.811 ** | −0.480 | 0.804 ** | 0.854 ** |
| Mean ST at the silking stage | 5 | 0.094 | −0.149 | −0.217 | −0.857 ** | 0.929 ** | 0.924 ** |
| | 10 | −0.058 | −0.130 | −0.240 | −0.805 ** | 0.677 * | 0.581 |
| | 15 | −0.485 | 0.379 | 0.278 | −0.947 ** | 0.845 ** | 0.788 * |
| | 20 | 0.715 * | −0.787 * | −0.786 * | −0.509 | 0.821 ** | 0.863 ** |
| | 25 | 0.593 | −0.410 | −0.237 | −0.396 | 0.665 | 0.720 * |
| Mean ST at the milk stage | 5 | −0.050 | −0.208 | −0.390 | −0.812 ** | 0.918 ** | 0.893 ** |
| | 10 | 0.181 | −0.361 | −0.490 | −0.612 | 0.852 ** | 0.883 ** |
| | 15 | 0.210 | −0.298 | −0.330 | −0.214 | 0.386 | 0.401 |
| | 20 | 0.580 | −0.791 * | −0.871 ** | 0.058 | 0.358 | 0.452 |
| | 25 | −0.338 | 0.001 | −0.196 | 0.460 | 0.018 | 0.135 |
| Mean ST at the physiological maturity stage | 5 | 0.875 ** | −0.844 ** | −0.776 * | −0.775 * | 0.834 ** | 0.805 ** |
| | 10 | 0.770 * | −0.926 ** | −0.959 ** | −0.358 | 0.647 | 0.686 * |
| | 15 | 0.253 | −0.559 | −0.710 * | −0.728 * | 0.882 ** | 0.875 ** |
| | 20 | 0.780 * | −0.925 ** | −0.942 ** | −0.064 | 0.521 | 0.624 |
| | 25 | 0.534 | −0.676 * | −0.713 * | 0.619 | −0.162 | −0.043 |

[a] SWS ( maize growth period(MGP)), mean soil water storage in the 0–200 cm soil layer throughout the maize growth period; ST (MGP), mean soil temperature in the 5–25 cm soil layer throughout the entire maize growth period. [b] Correlation coefficients: *, significant at $p \leq 0.05$; **, significant at $p \leq 0.01$.

Compared to the DRWP treatment, the reduction in grain and biomass yields with DRBP was less than that of MSSM, and grain and biomass yields of MSSM were significantly reduced by 21.77% and 18.47% in 2017, and by 15.80% and 6.49% in 2018, respectively (Table 1). Averaged across mulching methods, silage, grain, and biomass yields were significantly ($p < 0.001$) 21.86%, 18.14%, and 25.69% greater, respectively, in 2018 than 2017 (Table 1). Grain and biomass yields were significant negatively correlated with mean SWS in the 0–200 cm soil layer throughout the maize growing season, but grain

and biomass yields were significantly positively correlated with mean soil temperature in the 0–25 cm soil layer and at the 5, 10, 15, and 20 cm depths throughout the maize growing season (Table 2). Soil temperature at the 5, 10, 15, 20, and 25 cm depths at V2 and V6 was significantly positively correlated with grain and biomass yields in 2017 and 2018, but soil temperature at the 5, 10, 15, 20, and 25 cm depths at R1 and the 5 and 10 cm depths at V6 was significantly positively correlated with grain and biomass yields in 2018. Soil temperature at the 5, 10, 15, 20, and 25 cm depths at R6 was significantly negatively correlated with grain and biomass yields in 2017; in contrast, soil temperature at the 5, 10, and 15 cm depths at R6 was significantly negatively correlated with grain and biomass yields in 2018 (Table 2).

Across the two years, grain yield was significantly positively correlated with thousand kernel weight, followed by kernels per ear and ears per hectare, and thousand kernel weight was significantly positive correlated with kernels per ear and ears per hectare (Table 3). Maize ears per hectare and thousand kernel weight were significantly lower with the MSSM and DRBP treatments than the DRWP treatment in 2017 and 2018. Compared to the DRWP treatment, ears per hectare of the DRBP treatment was reduced by 1.48% and 1.45% ears ha$^{-1}$ in 2017 and 2018, and that of the MSSM treatment was significantly reduced by 1.29% and 2.40% in 2017 and 2018, respectively. Compared to the DRWP treatment, thousand kernel weight of the DRBP treatment was reduced by 2.48% and 3.75% in 2017 and 2018, and that of the MSSM treatment was significantly reduced by 17.62% and 11.16% in 2017 and 2018, respectively. Kernels per ear with the MSSM treatment was significantly lower than that with the DRWP and DRBP treatments in 2017 and 2018. Compared to the DRWP treatment, kernels per ear of the DRBP treatment was significantly increased by 3.32% in 2017, but was not significantly different in 2018, and that of the MSSM treatment was significantly reduced by 3.86% and 2.83% in 2017 and 2018, respectively. Averaged across mulching methods, kernels per ear and thousand kernel weight were significantly ($p < 0.001$) 13.06% and 4.81% greater, respectively, in 2018 compared to 2017.

**Table 3.** Pearson's correlation coefficients for correlations among maize grain yield and its components, across mulching methods in 2017 and 2018.

| Year | Dependent Variable | Grain Yield | Ears ha$^{-1}$ | Kernels ear$^{-1}$ |
|---|---|---|---|---|
| | Ears ha$^{-1}$ | 0.414 | | |
| 2017 | Kernels ear$^{-1}$ | 0.867 ** | −0.069 | |
| | Thousand kernel weight (g) | 0.995 ** | 0.495 | 0.813 ** |
| | Ears ha$^{-1}$ | 0.905 ** | | |
| 2018 | Kernels ear$^{-1}$ | 0.850 ** | 0.598 | |
| | Thousand kernel weight (g) | 0.979 ** | 0.935 ** | 0.792 * |

Correlation coefficients: *, significant at $p \leq 0.05$; **, significant at $p \leq 0.01$.

### 3.6. Economic Benefits of Silage, Grain, and Biomass Yields

Total investment cost was the same for the DRWP and DRBP treatments, and was least with the MSSM treatment (Table 4). The interaction between mulching treatment and year significantly influenced revenue ($p < 0.001$) and net economic return from silage ($p < 0.001$), and revenue ($p = 0.002$) and net economic return from grain ($p = 0.002$). Revenue and net economic return from silage with the MSSM treatment were significantly higher than that with the DRWP and DRBP treatments in 2017 and 2018. Greater revenue and net economic return from grain occurred in 2018 than 2017 ($p < 0.001$). However, revenue and net economic return from grain with the MSSM treatment were significantly lower than those with the DRWP and DRBP treatments in 2017 and 2018. Revenue and net economic return from grain were greater in 2018 than 2017 ($p < 0.05$). Compared to the DRWP treatment, revenue and net economic return from silage of the DRBP treatment was reduced by 3.11% and 4.39% in 2017, and by 4.01% and 5.26% in 2018, and those with the MSSM treatment were significantly increased by 6.55 and 25.49% in 2017, and by 4.15% and 17.67% in 2018, respectively. Averaged across mulching methods, revenue and net economic return from silage were significantly ($p < 0.001$) 21.86% and 29.19%

greater, respectively, in 2018 than in 2017. Revenue and net economic return from grain of the DRBP treatment were significantly reduced by 4.07% and 6.63% in 2018, and those with the MSSM treatment were significantly reduced by 21.77% and 7.36% in 2017, and by 15.80% and 1.06% in 2018, respectively. Averaged across mulching methods, revenue and net economic return from grain were significantly ($p < 0.001$) 18.14% and 31.49% greater, respectively, in 2018 than in 2017.

**Table 4.** Total investment cost (TIC, Yuan ha$^{-1}$), revenue from silage (RS, Yuan ha$^{-1}$), net economic return from silage (NERFS, Yuan ha$^{-1}$), ratio of total investment cost to revenue from silage (TIC/RS), revenue from grain (RG, Yuan ha$^{-1}$), net economic return from grain (NERFG, Yuan ha$^{-1}$), and ratio of total investment cost to revenue from grain (TIC/RG) as affected by mulching method in 2017 and 2018.

| Year (Y) | Mulching Method (M) [a] | Dependent Variable | | | | | | |
|---|---|---|---|---|---|---|---|---|
| | | TIC | RS | NERFS | TIC/RS | RG | NERFG | TIC/RG |
| 2017 | DRWP | 8016 | 27,421b [b] | 19,405b | 0.292b | 17,771a | 9755a | 0.451b |
| | DRBP | 8016 | 26,569c | 18,553c | 0.302a | 17,627b | 9611b | 0.455a |
| | MSSM | 4866 | 29,218a | 24,352a | 0.167c | 13,903c | 9037c | 0.350c |
| 2018 | DRWP | 8016 | 33,784b | 25,768b | 0.237b | 20,792a | 12,776a | 0.386b |
| | DRBP | 8016 | 32,428c | 24,412c | 0.247a | 19,945b | 11,929b | 0.402a |
| | MSSM | 4866 | 35,187a | 30,321a | 0.138c | 17,507c | 12,641a | 0.278c |
| *p* > *F* | M | — | <0.001 | <0.001 | <0.001 | <0.001 | <0.001 | 0.002 |
| | Y | — | <0.001 | <0.001 | <0.001 | <0.001 | <0.001 | <0.001 |
| | M * Y | — | <0.001 | <0.001 | <0.001 | 0.002 | 0.002 | <0.001 |

[a] DRWP, double ridge-furrow fully mulched soil with white plastic film; DRBP, double ridge-furrow fully mulched soil with black plastic film; MSSM, maize straw strip mulching with alternating strips of mulched and non-mulched soil and maize planted in non-mulched strips. [b] Within a row for a given year, means followed by different letters are significantly different ($p \leq 0.05$).

The interaction between mulching treatment and year significantly ($p < 0.001$) influenced the ratio of total investment cost to revenue from grain. Compared to the DRWP treatment, the ratio of total investment cost to revenue from silage with the DRBP treatment was not significantly different and that with the MSSM treatment was reduced by 12.58% and 9.90% in 2017 and 2018 (Table 4). Averaged across mulching methods, the ratio of total investment cost to revenue from silage was significantly ($p < 0.001$) 18.12% lower in 2018 than 2017; the ratio of total investment cost to revenue from grain with the DRBP treatment was not significantly different, and that with the MSSM treatment was reduced by 10.11% and 10.76% in 2017 and 2018, respectively (Table 4). Averaged across mulching methods, the ratio of total investment cost to revenue from grain was significantly ($p < 0.001$) 17.88% lower in 2017 than in 2018.

### 3.7. Agricultural Ecological Environment before and after Straw Strip Mulching Crops in RainFed Agricultural Regions

Figure S5 (A1–A4) shows pollution of the agricultural environment due to open field burning of residual plastic film in surface soil and maize straw. Figure S5 (A5–A6) reveals that soil moisture is suitable for crop development under straw strip mulching and double ridge mulching. However, a difference is that 10–12 earthworms are present within an area of 1.0 m × 0.5 m in the 0–10 cm soil layer under straw strip mulching, compared to 0–2 earthworms under plastic film mulching. Figure S5 (A9–A14) shows winter wheat, potato, and spring maize production under straw strip mulching at Tongwei, Gansu Province, China in 2017 and 2018. Figure S5 (A15–A16) is of the field after maize harvest under double ridge mulching and straw strip mulching, and shows that plastic film mulching film is difficult to remove, while straw mulch can continue to cover the soil environment.

## 4. Discussion

### 4.1. Straw Strip Mulching Affects the Temporal and Spatial Dynamics of Soil Temperature

Maize is a thermophilic C4 crop, and suitable air and soil temperatures are key factors that affect its growth and development [61]. Meeting the temperature requirements at each crop stage can coordinate and promote maize growth and development to obtain high yield and net economic return. Our results show that the MSSM treatment significantly reduced mean soil temperature of the 5–25 cm soil layer by 1.27 and 2.03 °C throughout the growing season compared to that of the DRBP and DRWP treatments in 2017 and 2018, which is consistent with the results of Lu et al. [62], Chen et al. [52], and Chang et al. [53], respectively. In 2017, the largest difference in mean soil temperature among treatments was at the 25 cm soil depth, with the MSSM and DRBP treatments being 2.57 and 1.54 °C lower compared to the DRWP treatment, respectively. The largest difference between the MSSM and DRBP treatments in 2018 was in the 5 cm and 25 cm soil layers, which were 2.75 and 1.73 °C lower than that of DRWP, respectively. The largest soil temperature difference appeared at the V2 stage in 2017 and 2018. At the V2 stage, compared to the DRWP treatment, soil temperature under the MSSP treatment was 7.45 °C less at the 15 cm soil depth in 2017, and 4.82 °C less at the 10 cm soil depth in 2018. Li [63] reported that soil temperature at a depth of 10 cm of alternating ridges and furrows where only the ridges were mulched with plastic film was significantly increased by 0.5–4.5 °C compared to no mulching under low air temperatures during the early growth period of maize. The present study shows that in 2017, compared to the DRWP treatment, mean soil temperature of the 5–25 cm soil layer of the MSSM treatment was significantly decreased during the vegetative growth stages by 5.32–6.62 °C and increased during the reproductive growth stages by 0.59–1.16 °C, and that with the DRBP treatment was significantly decreased during the vegetative growth stages by 2.35–2.52 °C and increased during the reproductive growth stages by 0.50–0.50 °C. In 2018, compared to the DRWP treatment, mean soil temperature of the 5–25 cm soil layer of the DRBP and MSSM treatments in 2018 was significantly reduced throughout the growing season by 0.35–1.36 and 0.76–4.21 °C, respectively. These results are consistent with Lu et al. [64], who reported that compared to white plastic film mulching, black plastic film mulching reduced soil temperature of the 0–15 cm soil layer in the daytime by 0.8 °C, prolonged the grain-filling period of maize, and increased dry matter accumulation per plant after silking. Black plastic film mulching has also been shown to reduce the diurnal amplitude of soil temperature [65]. Soil temperature at a depth of 10 cm of alternating ridges and furrows where only the ridges were mulched with plastic film was significantly increased by 0.5–4.5 °C compared to no mulching under low air temperatures during the early growth period of maize [63]. Previous studies also reported that plastic mulching often increases soil temperature, whereas straw mulching has less of a soil warming effect for maize under low air temperatures [66,67]. This is in agreement with Chang et al. [53], who reported that straw strip mulching reduced soil temperature during the early growth of potato. The temperature reduction under straw strip mulching in some growth stages can have a positive value for the growth and development of potato [53]. Lower soil temperature during the grain-filling period is conducive to maize, as it results in smaller diurnal temperature fluctuations, leading to lower respiration and enhanced dry matter accumulation [68,69].

The mechanism of soil temperature decrease under straw strip mulching is due to the ability of the mulching material to reflect and transmit solar energy [70]. The other mechanism of soil temperature reduction under straw strip mulching was that, by regulating water with temperature, the vigorous growth period and peak water consumption of crops will move to the precipitation concentrated season, realizing water–temperature coupling. This is in agreement with Li et al. [53] and Chang et al. [57], who reported that straw mulching advanced soil moisture and decreased soil temperature in some growth stages of maize and potato, and the optimized soil microenvironment promoted crop reproductive growth and development. In the present study, compared with the DRWP treatment, the V2, V6, R2, and R6 stages of the MSSM treatment were delayed by 7, 3, 3 and 3 days in 2017, and by 7, 2, 3 and 3 days in 2018, respectively. This was due to the total mean daily

temperature accumulation under the MSSM treatment being 118.06 °C greater than that of the DRWP treatment in 2017, although there was no difference with the DRWP treatment in 2018. Compared to the DRWP treatment, soil temperature under the DRBP and MSSM treatments was warmer at 52% and 56% of all measurement points in 2017, and cooler at 88% and 100% of all measurement points in 2018. Soil temperature accumulation under the DRBP and MSSM treatments increased at some growth stages compared to the DRWP treatment, along with maize development in 2017 and 2018. Changes in thermal conditions have a great impact on crop production [71], and the MSSM treatment improved the thermal conditions for maize production, despite its reduction in mean soil temperature in the 0–25 cm soil layer throughout the growing season.

### 4.2. Straw Strip Mulching Increased Soil Water Storage

Soil water storage is one of the most important factors influencing crop production [72], especially in regions with scarce precipitation, uneven distribution of precipitation, and high evaporation. Previous studies have shown that straw mulching and alternating ridges and furrows where only the ridges are mulched with plastic film can increase soil water content compared to no mulching under dryland conditions [52,66,73] and promote water infiltration into soil [74]. In the present study, compared with the DRWP treatment, the DRBP and MSSM treatments significantly increased mean SWS in the 0–200 cm soil layer by 14.35 and 24.10 mm in 2017, and by 4.21 and 20.00 mm in 2018, respectively. These results are consistent with previous studies [67]. Maize straw enhanced the rate of infiltration of rainwater and was more effective at increasing SWS than double ridge mulching with black plastic film [38,39,52]. However, the level of soil moisture depends on soil moisture conservation, infiltration, and evapotranspiration. The mechanism of increased SWS in the 0–200 cm soil layer under straw strip mulching was: (1) With MSSM, there were alternating strips of mulched (0.5 m wide) and non-mulched (0.7 m wide) soil; therefore, straw strip mulching only covered 46% of the soil. Although the effect of soil moisture preservation with MSSM is not as good as that with plastic film mulching, the infiltration rate of precipitation during maize growth is higher than that of plastic film mulching. (2) Maize straw flocculent pith is a good material for water absorption and water conservation. Its thicker wax layer has a better effect on water conservation than other crop straws. (3) Straw strip mulching significantly reduces ground temperature, surface evaporation, and plant transpiration intensity compared to non-mulched bare soil, but is similar or higher than that of plastic film mulching, which is conducive to reducing luxury water consumption and high-efficiency water consumption. This is the main reason why the soil moisture content of straw strip mulching was similar or higher to that of plastic film mulching.

### 4.3. Straw Strip Mulching Increased the Silage Yield, But Grain Yield Was Opposite

Endogenous soil hydrothermal conditions determine crop production [75,76]. In this study, greater effective precipitation at the V6 and R1 stages due to higher SWS under the MSSM treatment improved soil hydrothermal conditions for maize growth on the side of the furrows with the MSSM treatment, and lead to significantly greater aboveground fresh weight per plant at 143 days after planting. Aboveground plant water content and silage yield of MSSM were significantly higher than those of the DRWP and DRBP treatments at silage harvest. At the time of silage harvest, maize phenological development was more advanced for the DRWP and DRBP treatments compared to the MSSM treatment, as indicated by the X2 silage harvest parameter (Table 1). Filya [59] reported that whole plant maize silage has more lactic acid, ethanol, weight loss, yeast, and molds at the early dent maturity stage, and that optimum fermentable nutrients were present at a slightly later dent stage (two-thirds kernel milk line) [59]. However, grain yield under the MSSM treatment was lower than that of the DRWP and DRBP treatments at the R6 stage. This is in agreement with Afyuni et al. [77], who reported that silage yield was positively correlated with soil water content. Li et al. [78] found that maize silage yield under stubble mulching without tillage was 9%–11% greater and water productivity was increased by 11.7%–14.8% when compared with tillage and stubble burning without tillage treatments,

which led to higher soil water content in the 0–20 cm soil layer in a drier year and in the 0–120 cm soil layer in a wetter year.

Previous studies have shown that plastic film mulching improves soil conditions and enhances the growth and productivity of maize in rainfed areas when compared with a non-mulched control [66,79,80]. Plastic film mulching can reduce soil evaporation, increase plant transpiration, and result in similar or greater evapotranspiration compared to no mulching [81], thereby enhancing crop canopy development [66]. The results of this study are consistent with those of previous studies showing that plastic mulch can improve WUE and increase maize yield in rainfed areas [80,82,83]. Grain yield and development had different sensitivity to changes in soil hydrothermal conditions based on mulching methods and rainfall years; grain yield of maize under plastic ridge—straw mulch and straw mulch treatments were similar and significantly higher than that of the non-mulched control treatment in 2007 (302 mm precipitation, dry year), but the straw mulch treatment had significantly lower grain yield than plastic ridge—straw mulch and non-mulched control treatments in 2008 (340 mm precipitation, wet year) [84].

In this study, grain yield reduction under the MSSM treatment was likely due to the straw mulch treatment having lower or similar yield than plastic film mulching treatments, which led to significant differences in soil micro-conditions under different rainfed regions with different precipitation and accumulated temperature, and soil temperature is the key factor that depends on the balance between the ability of the mulching material to reflect and transmit solar energy [70]. In the present study, maize silage yield was greater under MSSM than the DRWP and DRBP, while the opposite result occurred for grain yield. This can be attributed the following: (1) lower soil temperature and maize development with the MSSM treatment at the seedlings stage, but after the V6 stage the temperature fluctuation in each soil layer was small, and frequency of precipitation events >15 mm was greater from V7 to R1 than at other stages. After the V7 stage, maize grows rapidly, the crop canopy closes, and there is little difference in soil evapotranspiration among mulching treatments. However, after heavy rainfall, there is greater infiltration with straw mulching than plastic film mulching, which is why the SWC of straw strip mulching is higher than that of plastic film mulching. The optimized soil water and heat environment promoted later vegetative growth and prolonged the development time of the mid-dent stage, and improvements in maize silage and grain yields were likely associated with enhanced crop root growth well under superior hydrothermal conditions. (2) The control mechanism of realizing water–temperature coupling, pre-control, and post-promotion led to the MSSM treatment having similar or higher silage yield. Meanwhile, this will naturally reduce the pressure of centralized acquisition of plastic film mulched silage maize and straw mulched silage maize for silage maize acquisition enterprises, and straw mulched silage maize was greener at harvest than plastic film mulched silage maize. (3) Maize development was delayed under the MSSM treatment, which can be advantageous for timely harvest of silage maize but may not be well suited for grain development. Since grain yield of MSSM was lower than that of DRWP and DRBP in both study years, future research should identify an optimized suite of agronomic practices to increase grain yield of maize grown with MSSM.

### 4.4. Straw Strip Mulching Increased Net Economic Return from Silage Yield

In this study, net economic return from silage with the MSSM treatment was significantly higher by 28.31% and 20.85% in 2017 and 2018, respectively. In contrast, net economic return from grain with the MSSM treatment was lower by 6.67% in 2017 and higher by 2.34% in 2018, than those with the double ridge mulching system. Compared to the DRWP and DRBP treatments, the ratio of total investment cost to revenue from silage and grain for the MSSM treatment was reduced 22.72–43.9% in 2017 and by 29.41–42.91% in 2018. These results are similar to those of Pishgar Komleh et al. [27], who found that greater yield led to greater benefit to cost ratio [27]. Zhao et al. [21] showed that straw mulching decreased greenhouse gas emission intensity by 47% and 40%, and net economic return under straw mulching increased by 13% and 27%, compared to plastic mulching and ridge-furrow plastic mulching plus furrow seeding treatments, respectively [21]. Adding a high concentration maize

silage to feed can improve the economic benefits, sustainability, and carrying capacity of livestock production [85–87]. The high net economic return from maize silage in this study was due to the use of idle straw as a mulching material, with zero material cost. At the same time, many ecological livestock production systems, such as free range and organic production systems, can reduce input costs by adding silage maize as roughage to feed [87,88]. Therefore, straw strip mulching technology is a feasible choice for either silage or grain maize production in dryland agricultural regions.

*4.5. Straw Strip Mulching as an Alternative to Plastic Film Mulch in Maize Production in Rainfed Regions*

Crop straw and plastic film residue are on the rise with the increasing demand for high yield, and burning straw and plastic film has become one of the shortcut ways for dealing with straw and residual plastic film before new cropping seasons [89,90]. This is especially common for dealing with crop straw in rice production in India, Thailand and the Philippines [22]. Plastic film and crop straw residues in agricultural burns about 250 Tg of biomass from an annual biomass of open field burning [22], and air pollution has become one of the biggest threats to human health [23]. Furthermore, residual plastic film is difficult to degrade and has an inefficient recycling capacity, which is a major limiting factor for the sustainable development of agriculture because it reduces the homogeneity of soil and can seriously impede the movement of water and solutes in soil [18,20]. To address the environmental challenges posed by disposal of straw and plastic film, straw strip mulching could be developed for straw recycling in sustainable cropping. Straw strip mulching has significantly advanced winter wheat yield and decreased soil temperature when compared to traditional non-mulched flat planting, and it also increased potato yield and enhanced water use efficiency of dryland agriculture in northwestern China [51–53]. Straw mulch can also enhance soil aggregation, promote soil biological activity [42,43], stimulate plant growth and development [21], and improve crop productivity and quality [45]. Furthermore, governments should strengthen their guidance on straw strip mulching techniques and support farmers in the transformation of grain to silage production and their contribution to animal husbandry while improving the efficiency. The primary advantages to maize straw strip mulching include: (1) reduced air pollution from plastic film and crop straw burning in agricultural regions; (2) efficient utilization of waste straw resources; (3) enhanced soil quality, hydrothermal condition, and crop growth; (4) maize straw is a no cost environmentally friendly mulching material; (5) greater net economic return from silage than double ridge plastic film mulched systems. As a result, straw strip mulching could be a viable alternative to plastic mulching in maize production in the rainfed regions.

## 5. Conclusions

Although straw strip mulching showed lower maize grain yields than double ridge plastic film mulching systems, straw strip mulching increased SWS, decreased soil temperature in the 0–25 cm soil layer throughout the growing season, and reduced the fluctuation of soil temperature. In addition, it delayed the plant growth development accumulation days, but promoted aboveground fresh weight accumulation per plant at 143 days after planting under an enhanced soil hydrothermal condition relative to double ridge plastic film mulching systems. The improved soil hydrothermal condition likely contributed to greater maize silage yield and stay green of silage maize in the MSSM treatment, which could mitigate the contemporary problem of centralized acquisition of silage maize for silage maize acquisition enterprises. Farmers are highly interested in the net economic return from silage and grain maize under different mulching materials. Straw strip mulching outperformed double ridge mulching with plastic film in terms of net economic return from silage, which will be pivotal to promote its use, and thus reduce air pollution from burning residual crop straw and plastic film in rainfed regions of the Loess Plateau. Straw strip mulching as an alternative to plastic film mulching for maize production is a viable technique for sustainable agricultural development in semiarid rainfed areas.

**Supplementary Materials:** The following are available online at http://www.mdpi.com/2071-1050/12/15/6273/s1, **Figure S1.** Soil water storage in the 0–200 cm soil layer throughout the growing season as affected by mulching

method in 2017 and 2018. **Figure S2.** Maize growth development accumulation days after sowing at different stages as affected by mulching method in 2017 and 2018. **Figure S3.** The mean daily temperature accumulated temperature at different stages as affected by mulching method in 2017 and 2018. **Figure S4.** Aboveground fresh weight per plant at the days after seeds planting at different stages as affected by mulching method in 2017 and 2018. **Figure S5.** Agricultural ecological environment before and after straw strip mulching wheat, potato and maize production at Tongwei, Gansu Province, China in 2017 and 2018.

**Author Contributions:** Conceptualization, X.L., S.C., L.C. and C.H.; Data curation, H.C. and L.L.; Funding acquisition, S.C., H.C., L.C. and C.H.; Investigation, R.L., Y.C., Y.L., J.M. and L.L.; Methodology, S.C., R.L., Y.C. and J.M.; Project administration, S.C., H.C., L.C. and C.H.; Resources, S.C., L.C., C.H., R.L., Y.C., Y.L., J.M. and L.L.; Software, L.C. and Y.L.; Supervision, S.C.; Validation, S.C., J.A.C., H.C., L.C., C.H., R.L., Y.C., Y.L., J.M. and L.L.; Writing—original draft, X.L.; Writing—review and editing, J.A.C. All authors have read and agreed to the published version of the manuscript.

**Funding:** This study was funded by the National key Research and Development program of China, Grant/Award Number: 2018YFD0200401 and National Natural Science Foundation of China, Grant/Award Number: 31760373, 31760362, 31960380.

**Acknowledgments:** The authors are grateful for assistance with field work by the staff of the Tongwei Modern Dryland Circular Farming Experiment Station.

**Conflicts of Interest:** The authors declare no conflict of interest.

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
