# Peer review of "Maize Straw Strip Mulching as a Replacement for Plastic Film Mulching in Maize Production in a Semiarid Region"

_sustainability, doi:10.3390/su12156273_

Round 1
Reviewer 1 Report
Sustainability
REVIEWER COMMENTS
MS No. Sustainability-884756
Title: Maize straw strip mulching as a replacement for plastic film mulching in maize production in a semiarid region
COMMENTS
In the manuscript is argued on a study concerning the possible application of different mulching.
The objectives of the study were:
(1) the application of straw strip mulching technology in maize production to reduce the air pollution and environmental pollution coming from plastic film and crop straw residues burning in agricultural regions;
(2) the comparison of the effects of double ridge-furrow fully mulched soil with white plastic (DRWP), double ridge-furrow fully mulched soil with black plastic (DRBP) and maize straw strip mulching (MSSM) treatments on soil temperature and soil water storage;
(3) the evaluation of the effects of DRWP, DRBP and MSSM treatments on maize growth development accumulation at different stages, silage yield, grain yield, biomass yield, WUE of grain and biomass,
and (4) assess the economic performance of straw strip mulching compared with plastic mulching.
The objectives of the study are surely interesting from both agronomic and scientific point of view.
The manuscript is well structured and organized, according to the journal directions (Introduction; Material and methods; Results, Discussion, Conclusions).
In the reading of the text, I found it pretty well written and arranged in the description.
Nevertheless, I found some points for which I have some remarks, reported below.
In Material and Methods.
Was the urea (180 kg N ha-1) applied just in one solution? Please specify
L 176: “both covered with polyethylene film (0.01 mm diameter)”, please check
Line 185: “Figure 2017. and mid-November in 2018 …” something is wrong, please correct it.
Concerning the statistical analysis, the authors say "Two-way analysis of variance was conducted for all dependent variables, ... Treatment and year were considered fixed effects, while replication and interactions with replication were considered random effects.". I have to highlight that the model is inappropriate due to the fact that the year has not been considered as repeated measure. I have to suggest that the use of mixed model is appropriate for the analysis of the data. Alternatively, they can provide an adequate model to run in which the year is considered repeated measure.
Results
The trend of climatic factors in the two years seem to be different, but in all reported data (figures and tables) it has never highlighted the difference due to the time (year). This is a little bit strange fact, therefore in my opinion if the year will result significant in the statistical analysis, it will be necessary report the data differently. In fact, the authors report often in the text the evidence of the year effect even if in the tables and figures it is not distinguished.
So, I suggest that these improvements of the manuscript are needed for publish it on Sustainability.
Author Response
Dear Editor and Reviewers, Sustainability
Thank you very much for your letter on 22 July. 2020 regarding manuscript (ID: sustainability-884756) entitled “Maize straw strip mulching as a replacement for plastic film mulching in maize production in a semiarid region’’. We are grateful to you and the reviewers for the and thoughtful suggestions for improving this manuscript. Based on those comments and suggestions, we carefully revised the original manuscript. We hope the revised manuscript will meet your journal’s standards for publication. However, we are open to addressing further questions or suggestions that may arise from this revision.
Thank you for your time and help with this manuscript. Our detailed point-by-point responses to the editor’s and reviewers’ comments are below. Revisions to original manuscript are shown with track changes. The line numbers in the editor’s and reviewers’ comments are those in the original manuscript reviewed by the reviewers. The line numbers in our author responses are those in the revised manuscript with track changes showing.
COMMENTS
In the manuscript is argued on a study concerning the possible application of different mulching.
The objectives of the study were:
(1) the application of straw strip mulching technology in maize production to reduce the air pollution and environmental pollution coming from plastic film and crop straw residues burning in agricultural regions;
(2) the comparison of the effects of double ridge-furrow fully mulched soil with white plastic (DRWP), double ridge-furrow fully mulched soil with black plastic (DRBP) and maize straw strip mulching (MSSM) treatments on soil temperature and soil water storage;
(3) the evaluation of the effects of DRWP, DRBP and MSSM treatments on maize growth development accumulation at different stages, silage yield, grain yield, biomass yield, WUE of grain and biomass,
and (4) assess the economic performance of straw strip mulching compared with plastic mulching.
The objectives of the study are surely interesting from both agronomic and scientific point of view.
Authors’ response: It's our pleasure that you are interested in our team’s research, and thank you very much for your support and approval!
The manuscript is well structured and organized, according to the journal directions (Introduction; Material and methods; Results, Discussion, Conclusions).
Authors’ response: It's our pleasure that you are interested in our team’s research, and thank you very much for your support and approval!
In the reading of the text, I found it pretty well written and arranged in the description. Authors’ response: It's our pleasure that you are interested in our team’s research, and thank you very much for your support and approval!
Nevertheless, I found some points for which I have some remarks, reported below.
In Material and Methods.
L 176: Was the urea (180 kg N ha-1) applied just in one solution? Please specify.
Authors’ response: Thank you for this question, which we have followed (Lines: 180-184).
L 176: “both covered with polyethylene film (0.01 mm diameter)”, please check.
Authors’ response: Thank you for this question, which we have followed (Lines: 193-195).
Line 185: “Figure 2017. and mid-November in 2018 …” something is wrong, please correct it. Authors’ response: Thank you for this question, which we have followed (Lines: 304-305).
Concerning the statistical analysis, the authors say "Two-way analysis of variance was conducted for all dependent variables, Treatment and year were considered fixed effects, while replication and interactions with replication were considered random effects." I have to highlight that the model is inappropriate due to the fact that the year has not been considered as repeated measure. I have to suggest that the use of mixed model is appropriate for the analysis of the data. Alternatively, they can provide an adequate model to run in which the year is considered repeated measure.
Results
The trend of climatic factors in the two years seem to be different, but in all reported data (figures and tables) it has never highlighted the difference due to the time (year). This is a little bit strange fact, therefore in my opinion if the year will result significant in the statistical analysis, it will be necessary report the data differently. In fact, the authors report often in the text the evidence of the year effect even if in the tables and figures it is not distinguished.
Authors’ response: Thank you for this question, which we have followed (Lines: 434-436). 1. The text for the methods of data analysis that is listed in our manuscript is analysis of variance. 2. In Table 1 and 3 in our manuscript , we delete the rows that begin with "ANOVA F Pr." for 2017 and 2018. Only show this information for 2017-2018. Also, don't show means for 2017-2018 (this is because the interaction was significant, so need to show means separately by year). Hope these updates could improvements for our manuscript.
So, I suggest that these improvements of the manuscript are needed for publish it on Sustainability.
Authors’ response: It's our pleasure that you are interested in our team’s research, and thank you very much for your support and approval!
Reviewer 2 Report
In my opinion the study is of a good quality enough. Yet, the introduction mist be better expoained, the methodology's reasons should be motivated deeper, the implication cancan be improved. Other recent and important references of several context must be considered, for example: Boccia, Di Donato, Covino, Poli (2019) in Journal of Cleaner Production n. 227.
Author Response
Dear Editor and Reviewers, Sustainability
Thank you very much for your letter on 22 July. 2020 regarding manuscript (ID: sustainability-884756) entitled “Maize straw strip mulching as a replacement for plastic film mulching in maize production in a semiarid region’’. We are grateful to you and the reviewers for the and thoughtful suggestions for improving this manuscript. Based on those comments and suggestions, we carefully revised the original manuscript. We hope the revised manuscript will meet your journal’s standards for publication. However, we are open to addressing further questions or suggestions that may arise from this revision.
Thank you for your time and help with this manuscript. Our detailed point-by-point responses to the editor’s and reviewers’ comments are below. Revisions to original manuscript are shown with track changes. The line numbers in the editor’s and reviewers’ comments are those in the original manuscript reviewed by the reviewers. The line numbers in our author responses are those in the revised manuscript with track changes showing.
Comments and Suggestions for Authors
In my opinion the study is of a good quality enough. Yet, the introduction mist be better expoained, the methodology's reasons should be motivated deeper, the implication can be improved. Other recent and important references of several context must be considered, for example: Boccia, Di Donato, Covino, Poli (2019) in Journal of Cleaner Production n. 227.
Authors’ response: Thank you very much for your kind suggestions and good references, which we have followed (Lines: 89-90, Lines: 97-100, Lines: 105-109, Lines: 368-371, Lines: 994-1000).
Reviewer 3 Report
Lan et al. reported an interesting study regarding the production of maize using straw strip mulching as a replacement for plastic film mulching in a semiarid region.
The manuscript well fit with aim of the Journal and only some issues should be addressed:
Line 158. Please report the complete description of the soil (chemical and physical characterization) that is fundamental for this study
Line 164. When the fertilizers were applied?
Which plastic film was used? Please add provider
Line 176. Diameter?
I suggest to include some parameters related to the silage and grain quality, that can valorize the manuscript
Lines 139-140. This topic was not investigated in this manuscript
Author Response
Dear Editor and Reviewers, Sustainability
Thank you very much for your letter on 22 July. 2020 regarding manuscript (ID: sustainability-884756) entitled “Maize straw strip mulching as a replacement for plastic film mulching in maize production in a semiarid region’’. We are grateful to you and the reviewers for the and thoughtful suggestions for improving this manuscript. Based on those comments and suggestions, we carefully revised the original manuscript. We hope the revised manuscript will meet your journal’s standards for publication. However, we are open to addressing further questions or suggestions that may arise from this revision.
Thank you for your time and help with this manuscript. Our detailed point-by-point responses to the editor’s and reviewers’ comments are below. Revisions to original manuscript are shown with track changes. The line numbers in the editor’s and reviewers’ comments are those in the original manuscript reviewed by the reviewers. The line numbers in our author responses are those in the revised manuscript with track changes showing.
Comments and Suggestions for Authors
Lan et al. reported an interesting study regarding the production of maize using straw strip mulching as a replacement for plastic film mulching in a semiarid region.
Authors’ response: It's our pleasure that you are interested in our team’s research, and thank you very much for your support and approval!
The manuscript well fit with aim of the Journal and only some issues should be addressed:
Line 158. Please report the complete description of the soil (chemical and physical characterization) that is fundamental for this study.
Authors’ response: Thank you for this question, which we have followed (Lines: 174-177).
Line 164. When the fertilizers were applied?
Authors’ response: Thank you for this question, which we have followed (Lines: 180-184).
Which plastic film was used? Please add provider. Line 176. Diameter?
Authors’ response: Thank you for this question, which we have followed (Lines: 193-195).
I suggest to include some parameters related to the silage and grain quality, that can valorize the manuscript.
Authors’ response: Thank you for this question, which we have followed (Lines: 403-406, Lines: 627-638, Lines: 675-707, Lines: 723-725).
Lines 139-140. This topic was not investigated in this manuscript
Authors’ response: Thank you for this question, which we have followed (Lines: 775-808).
Authors’ response: Hope these updates could improvements for our manuscript.